

# The larvae of *Epigomphus jannyae* Belle, 1993 and *E. tumefactus* Calvert, 1903 (Insecta: Odonata: Gomphidae)

Rodolfo Novelo-Gutiérrez[1], Alonso Ramírez[2] and Débora Delgado[3]

[1] Red de Biodiversidad y Sistemática, Instituto de Ecología, A.C., Xalapa, Veracruz, Mexico
[2] Department of Environmental Science, University of Puerto Rico, San Juan, Puerto Rico
[3] Instituto Conmemorativo Gorgas de Estudios de la Salud, Panamá City, Panamá

## ABSTRACT

The taxonomic knowledge about immature stages of the insect order Odonata (dragonflies and damselflies) is rather limited in tropical America. Here, the larvae of *Epigomphus jannyae* Belle, 1993 and *E. tumefactus* Calvert, 1903 are described, figured, and compared with other described congeners. *E. jannyae* larva is characterized by 3rd antennomere 1.6 times longer than its widest part; ligula very poorly developed, with ten short, truncate teeth on middle; apical lobe of labial palp rounded and smooth. Lateral margins on abdominal segments (S5–9) serrated, lateral spines on S6–9 small and divergent; male epiproct with a pair of dorsal tubercles at basal 0.66; tips of cerci and paraprocts strongly divergent. The larva of *E. tumefactus* is characterized by 3rd antennomere 2.3 times longer than its widest part, ligula with 6–7 truncate teeth, apical lobe of labial palp acute and finely serrate. Lateral margins of S6–9 serrate, lateral spines on S7–9; male epiproct with a pair of dorsal tubercles at basal 0.50. Differences with other species were found in 3rd antennomere, lateral spines of S7–9, and the caudal appendages. *Epigomphus* larvae inhabit small, shallow creeks (1st order streams) where they live in fine benthic sediments. When mature, the larva leaves the water in shady places, climbing small rocks at the water's edge and metamorphosing horizontally on flat rocks. These new descriptions bring the total number of *Epigomphus* species with known larval stages to eight; only 28% of the species in this genus are known as larva.

# INTRODUCTION

Gomphidae is a cosmopolitan insect family of Odonata (dragonflies and damselflies) with over 980 species (*Dijkstra et al., 2013*). The family is diverse in streams and rivers, but also has several species that inhabit lentic environments (e.g., ponds, lakes) (*Garrison, von Ellenrieder & Louton, 2006*). Taxonomically, the adults in this group of dragonflies have been well-studied, although new species are frequently described. In contrast, and similar to most insect groups, the immature stages are poorly known. *Garrison, von Ellenrieder & Louton (2006)* report 255 species of gomphids for the New World, with only 177 of them known as larvae (69%). This lack of knowledge of the larval stages greatly limits our ability to understand the ecological role of insects in their ecosystems.

Corresponding author
Rodolfo Novelo-Gutiérrez,
rodolfo.novelo@inecol.mx

The genus *Epigomphus* is a poorly-known group of gomphids that inhabits small streams and rivers in the neotropics. *Epigomphus* comprises 28 species (*Garrison, von Ellenrieder & Louton, 2006*), but only six are known as larvae (larval descriptions in parentheses): *E. paludosus* Hagen *in* Selys, 1854 (*Martins, 1968*); *E. echeverrii* Brooks, 1989, *E. subobtusus* Selys, 1878, *E. subsimilis* Calvert, 1920 (all three by *Ramírez, 1996*); *E. hylaeus* Ris, 1918 (*Fleck, 2002*); and *E. crepidus* Kennedy, 1936 (*Novelo-Gutiérrez, Gómez-Anaya & Smith-Gómez, 2015*). In this paper, a detailed description and illustrations of the larvae of *E. jannyae* Belle and *E. tumefactus* Calvert are provided, based upon a specimen found emerging at the field, and reared larvae, respectively. Both species have restricted distributions and have been reported from a single country, *E. jannyae* is endemic to Panama and *E. tumefactus* to Costa Rica.

## MATERIALS & METHODS

*Epigomphus* larvae were collected in Panama and Costa Rica. Larvae were collected from the stream bottom and transported alive to the laboratory. Individuals emerging in the field were also collected. The male exuvia of *E. jannyae* was preserved in 96% ethanol while the teneral imago was maintained alive for a couple of days before it died and then preserved in ethanol. A couple of last instars (i.e., F–0) larvae of *E. tumefactus* were reared until emergence; another two F–0 larvae die and were preserved in ethanol. Emerged adults were identified using original descriptions and comparisons with identified specimens deposited at the Instituto de Ecología, A.C., Xalapa, Mexico.

Mandible nomenclature follows *Watson (1956)*; labium nomenclature follows *Corbet (1953)*. Photographs were obtained with a CANON PowerShot G10 digital camera mounted on a stereomicroscope ZEISS Stemi 2000-C. Measurements (in mm) were made with a calibrated ocular micrometer as follows: Head width, across compound eyes; total length, dorsally from anterior-most margin of labrum to tips of caudal appendages; abdomen length, ventrally from anterior margin of segment 1 to posterior margin of segment 10; hind femur, along midline of external surface. S1–10 = abdominal segments 1–10.

## RESULTS

### *Epigomphus jannyae* Belle

(Figs. 1, 2A, 3–4, 5A–5B, 6A–6B)

**Material.** One exuvia (male, emerged in the field). PANAMA: Panamá Oeste Province, Capira District, Altos de Campana Park, Sendero Panamá, stream (08°41.033N; 79°55.528 W, 835 masl), 25 April 2015 (1 male exuvia, emerged around noon), D. Delgado, A. Ramírez, R. Novelo leg. Deposited in the Colección Entomológica del Instituto de Ecología, A.C., Xalapa (IEXA).

**Description.** Medium-sized exuvia, body robust, antennae, legs, sides of thorax and abdomen setose, gently tapering caudad, body light yellow-brown and lacking any particular pattern (Fig. 1).

Head: Wider than long, even wider than pro- and mesothorax and basal abdomen (Fig. 1A). Labrum 0.6 mm long, mostly bare, anterior border with dense brush of golden

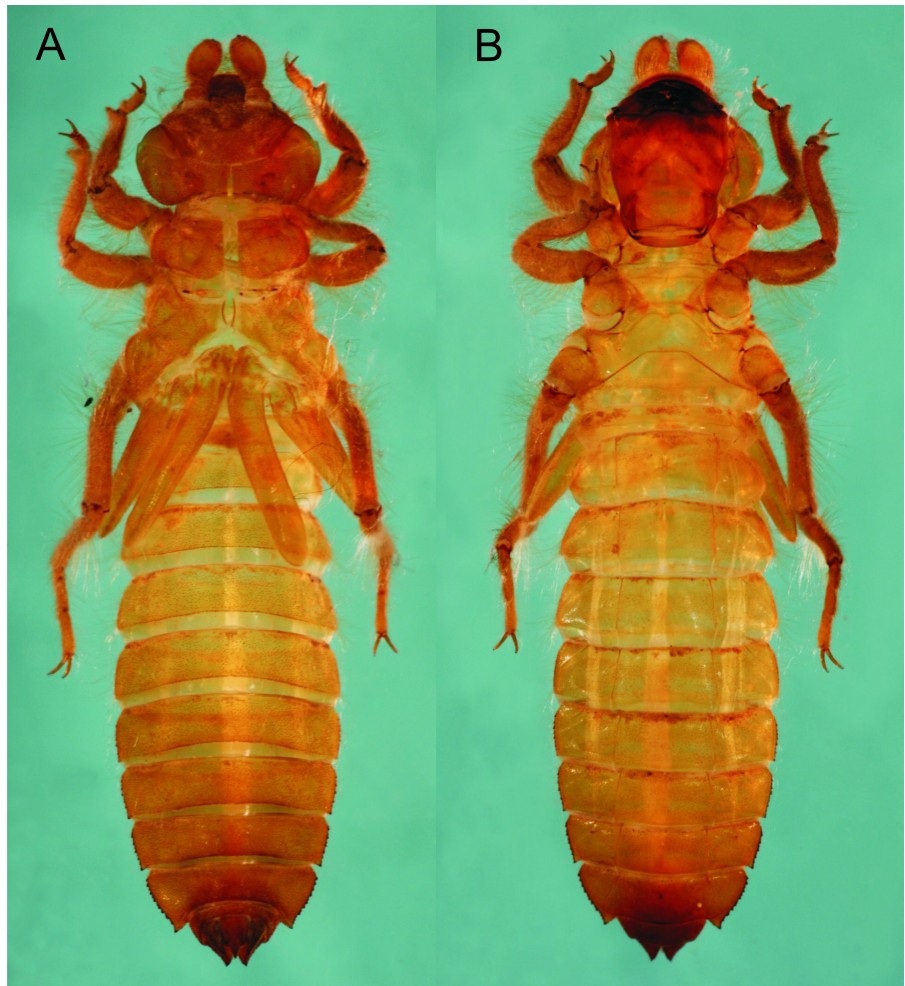

**Figure 1** *Epigomphus jannyae,* last stadium exuvia, (A) dorsal view, (B) ventral view.

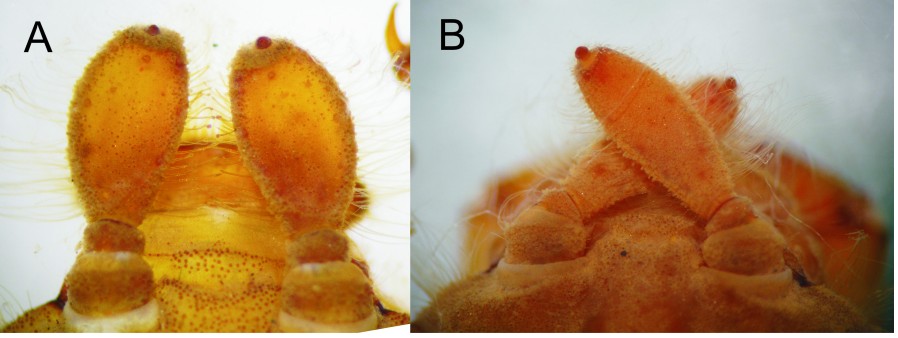

**Figure 2** **Details of the morphology of *Epigomphus* spp.** Dorsal view of antennae, (A) *E. jannyae*, (B) *E. tumefactus.*

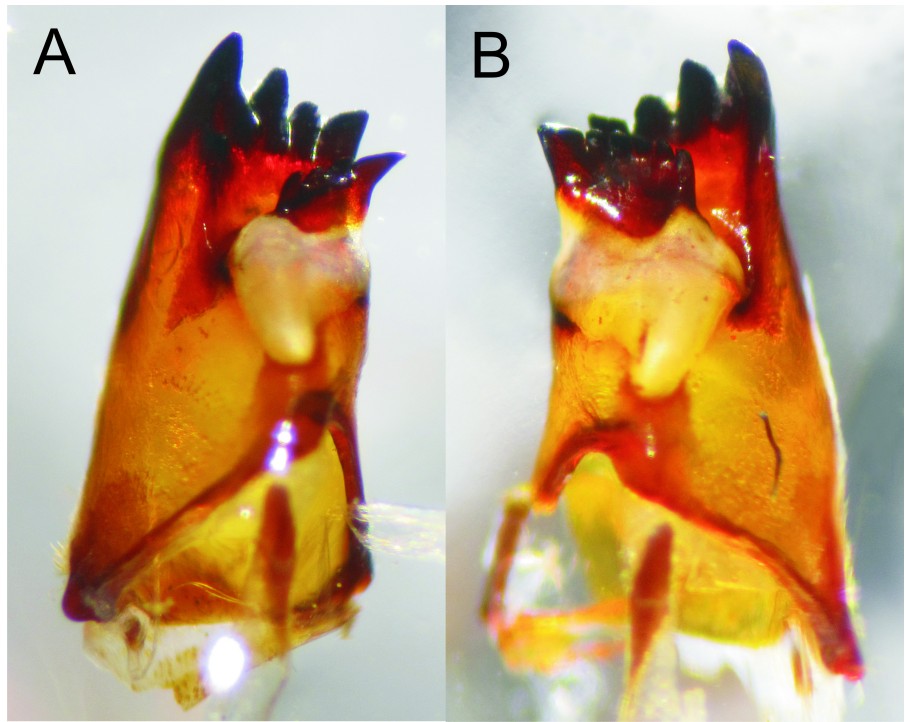

**Figure 3 Details of the morphology of *Epigomphus* spp.** Ventrointernal view of mandibles of *E. jannyae*, (A) right mandible, (B) left mandible.

setae, flattened ventrally; anteclypeus bare; postclypeus, frons, vertex and occiput finely granulose; a tuft of long, upturned, golden setae on fronto-lateral margins of frons; occiput mostly granulose with some bare, irregular areas and with long, golden setae on occipital lobes, anterior margin of frons concave. Antennae 4-segmented (Figs. 1A and 2A), covered with some sparse granuli on dorsum; scape globose, pedicel subglobose, both with abundant, small, scale-like setae on apical margin; 3rd antennomere largest, claviform, flattened dorso-ventrally, 1.6 times longer than its widest part, lateral and apical margins beset with abundant, small, scale-like setae, tightly packed on apical margin; dense rows of long, yellowish setae on lateral margins, those on external margins longest, those on internal margins stiff, with small, regularly spaced, circular structures close to the borders; 4th antennomere a small sphere; scape, pedicel and 3rd antennomere yellow brown, 4th antennomere reddish-brown, length proportions: 0.34, 0.15, 1.0, 0.06. Occipital lobes rounded, bulging; a well-developed longitudinal carina beset with small, stiff setae on each side of ventral surface of head (Fig. 1B). Mandibles (Fig. 3) with molar crest, formula: L 1234 0 a($m^{1,2,3,4}$b)/R 1234 y a($m^{1,2}$)b, in both mandibles tooth a > b. Maxillae: Galeolacinia (Fig. 4) with seven moderately incurved teeth; three dorsal teeth more or less of same length and robustness and four ventral teeth of different size, apical one largest; maxillary palp thick and robust. Ventral pad of hypopharynx pentagonal, whitish, soft, with a row of antero-ventral, subapical, long, stiff setae and a latero-basal triangular-shaped sclerite to each side of midline. Labium: Prementum-postmentum articulation reaching posterior margin of procoxae (Fig. 1B). Prementum reddish-yellow, subquadrate, as long as its

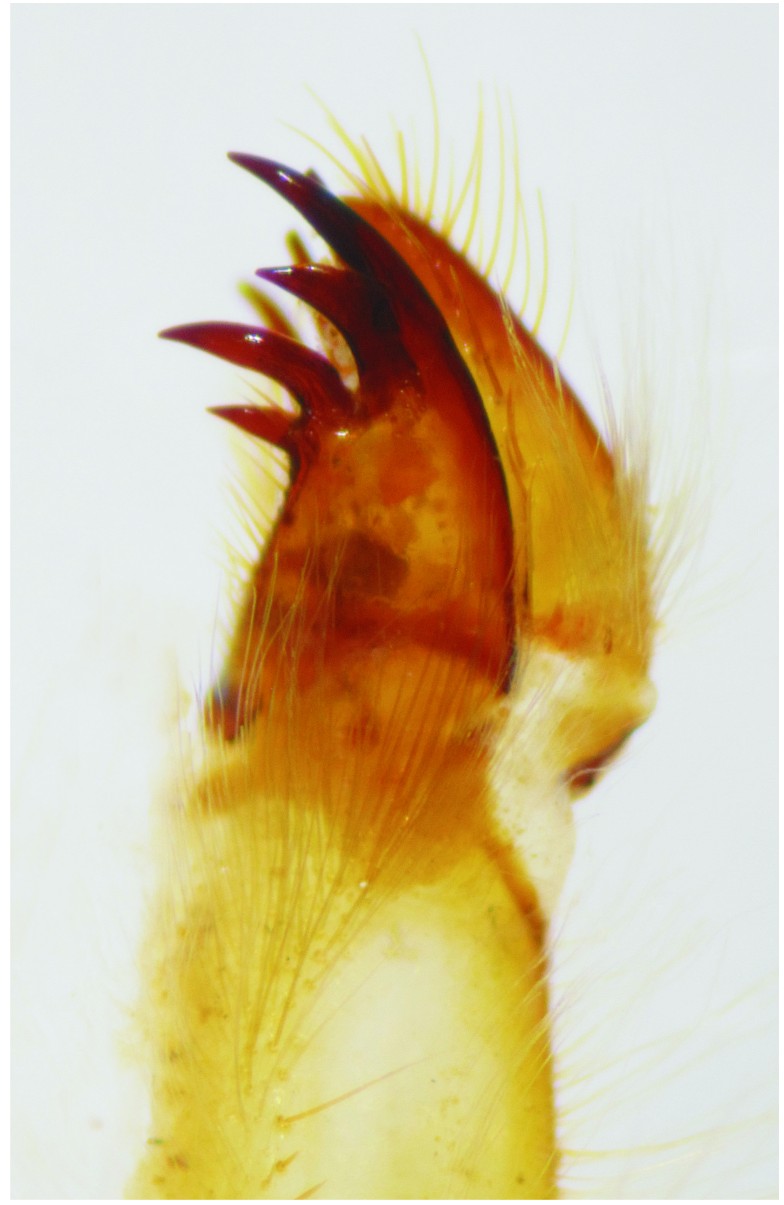

**Figure 4** **Details of the morphology of *Epigomphus* spp.** Maxilla's galeolacinia of *E. jannyae*, ventral view.

widest part (Fig. 5A); lateral margins with long whitish setae, slightly sinuate at apical 0.60, slightly sinuate and moderately convergent on basal 0.40; basal margin sinuate, without a longitudinal, central sulcus on ventral surface (Fig. 5A); a small, shallow, concavity just below ligula beset with some long setae (Fig. 5B). Ligula poorly developed (Fig. 5B), apical margin slightly convex, with a ventral row of 10 reddish-brown, short, truncate teeth on middle and a dorsal row of short, stout piliform setae. Labial palp stout (Figs. 5A–5B), reddish-brown, with abundant, long delicate, whitish setae on external surface; apical lobe stout, tip rounded and smooth, internal margin concave with 10–12 small teeth, the basal 6–7 teeth truncate and very close each other, remaining 4–5 teeth acutely pointed
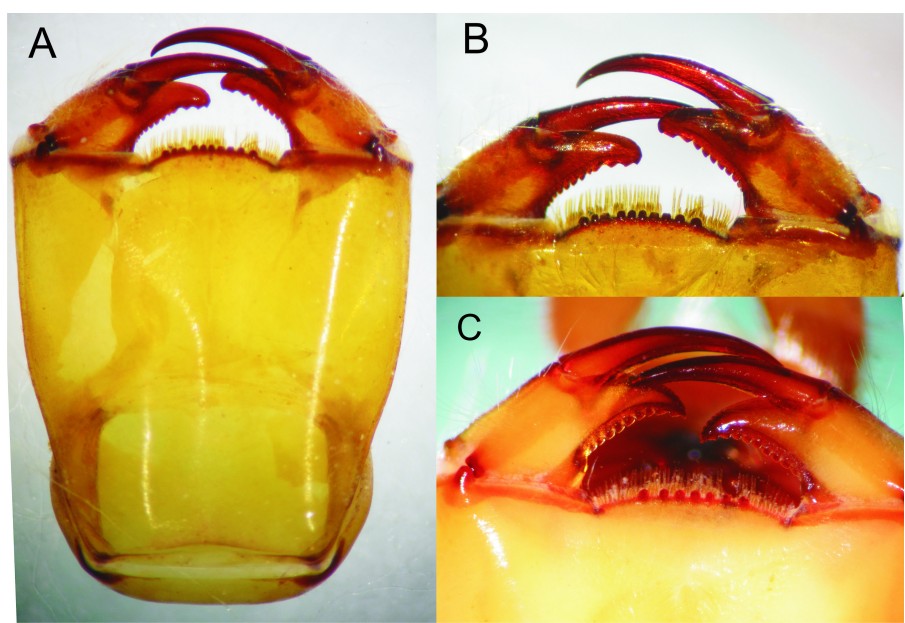

**Figure 5** **Details of the morphology of *Epigomphus* spp.** Prementum, ventral view: (A) *E. jannyae*, (B) Details of the ligula and labial palpi, *E. jannyae*, (C) Ídem but *E. tumefactus* (note acute tips of palpi).

and more separate each other; external margin gently convex and smooth; movable hook reddish-brown, almost as long as palp, sharp and moderately incurved.

Thorax: Pro- and mesothorax slightly narrower than head, setose on inferior border of pro-and mesopleura. Anterior margin of pronotum straight, lateral margins convex and bulging (similar to Fig. 7), posterior margin convex; a large, subquadrate glabrous area on each side of midline, remainder of pronotum granulose. Meso and metathorax granulose, some tufts of long, curled setae on inferior borders of meso- and metapleura, meso- and metaspiracles evident. Legs short (e.g., when fully extended, hind legs scarcely reaching posterior margin of abdominal segment 7), strongly setose (Fig. 1), with long, yellow-brown, delicate setae mainly on sides and shorter, stiff, reddish setae mainly on anterior surfaces of tibiae and tarsi; burrowing hooks moderately developed (Fig. 1); dorsal margins of metafemora the same length as metatibiae; tarsal formula 2-2-3, claws simple, with a pulvilliform empodium. Wing sheaths reaching posterior margin of S3, strongly divergent (Fig. 1A), mostly granulose, with abundant long, delicate setae on borders.

Abdomen: Yellow-brown; light reddish-brown on middle third of tergites 8–9, and the whole surface of S10 (Figs. 1 and 6A–6B); more or less spindle-shaped, ventral surface flat, dorsal surface convex, lacking dorsal protuberances, widest on S5–6; lateral margins of S1–7 with long, stiff yellowish setae, shorter on S8–10; lateral margins of S5–9 serrate (Figs. 1 and 6A), serrations on S5 very small and only visible in lateral and ventral views, serrations on S6–9 short and stout, dark reddish-brown; S6–9 ending in short (very small on S6 and only visible ventrally), triangular, divergent spines increasing in size and robustness posteriorly, those on S9 slightly upturned (Fig. 6A); tergites 2–10 granulose, with long, delicate setae mainly on the latero-dorsal margins, covering the lateral third of the sternite. Sternites

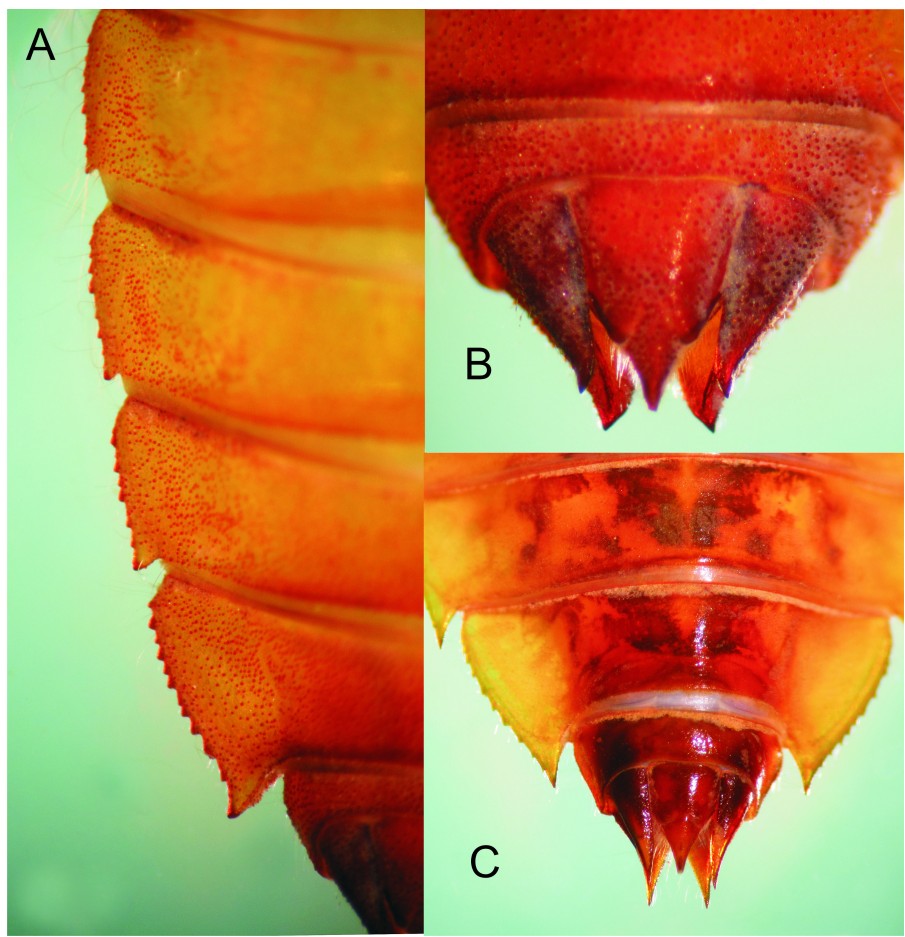

**Figure 6** **Details of the abdomen of *Epigomphus* spp.** (A) Lateral margins of *E. jannyae* segments 6–9 showing the serrate borders and apical spines; (B) Segment 10 and caudal appendages of *E. jannyae* showing the divergent tips of cerci and paraprocts; (C) Segments 8–10 of *E. tumefactus* showing acute lateral spines on 8–9 and sharply pointed caudal appendages.

following the same color pattern as tergites (Fig. 1B); sternites 3–8 divided into five plates, sternites 2 and 9 divided into three plates, ventral sutures parallel on 2–3, moderately divergent on 4–8, strongly divergent on 9. Male gonapophyses lacking. Caudal appendages pyramidal, obtusely pointed (Figs. 1 and 6B), epiproct and paraprocts reddish-brown, cerci dark reddish-brown, granulose on external surfaces, small, delicate, white setae on internal surfaces; male's epiproct with a pair of dorsal tubercles at basal 0.66 rounded apically and divergent; cerci and paraprocts, in dorsal view, mostly convergent except for the extreme tips of cerci, which are suddenly and strongly outcurved; dorsal and ventral margins of paraprocts ridged; paraprocts the shortest, epiproct the longest.

**Measurements.** Exuvia ($N = 1$): Total length 25.3; abdomen 16; maximum width of head 5.5; hind femur (lateral) 4.3; maximum width of abdomen (ventral) 6.5; epiproct 1.3, cerci 1.1, paraprocts 1.0; lateral spines on S6 0.10, on S7 0.12, on S8 0.20, on S9 0.4.

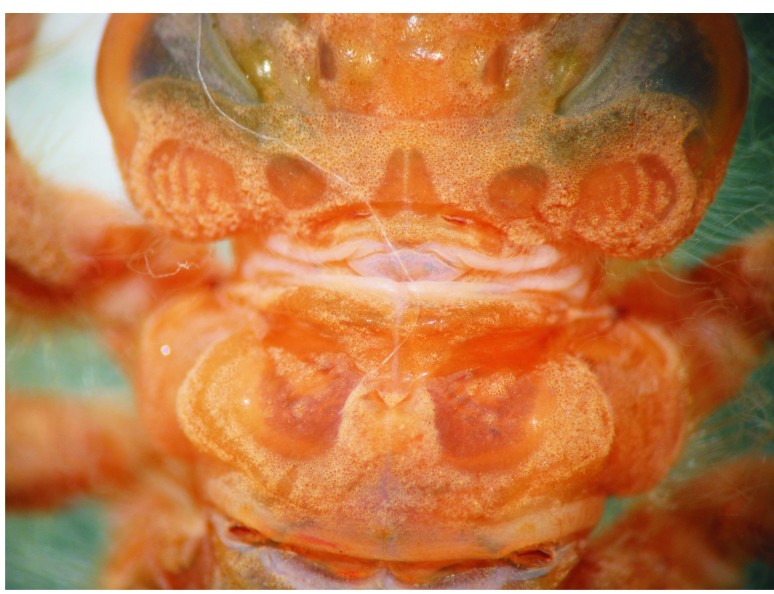

**Figure 7** **Details of the morphology of *Epigomphus* spp.** Occipital lobes of the head and prothorax, dorsal view, of *E. tumefactus*.

**Habitat.** A transforming individual was found in a small, shallow creek of 1st order, emerging on a small rock located close to the margin in a shady area. Emergence was horizontal, on a flat rock surface and lasted for more 40 min.

### *Epigomphus tumefactus* Calvert
(Figs. 2B, 5C, 6C, 7, 8)

**Material.** Two exuviae (male and female reared), 2 F–0 larvae (male, female). COSTA RICA: Heredia; Horquetas, Finca El Plástico, (10°17.0N; 84°01.60W, 700 masl), 25 March 1995 (4 F–0 specimens: 1 male and 1 female F–0 emerged, 1 male and 1 female larvae F–0 preserved in ethanol), A. Ramírez leg. Deposited in IEXA and in the Museo de Zoologia, Universidad de Cosa Rica (MZUCR).

**Description.** Medium-sized larvae, body robust, integument finely and abundantly granular, antennae, legs, sides of thorax and abdomen setose, gently tapering caudad, body yellow-brown lacking any defined pattern (Fig. 8).

Head: As described for *E. jannyae* except: Labrum 0.7 mm long; occiput mostly finely granulose with some bare, irregular areas (Fig. 7). Antennae yellow brown (Figs. 2B and 8), 3rd antennomere spindle-shaped, 2.3 times longer than its widest part, 4th antennomere a subconical rudiment, length proportions: 0.30, 0.14, 1.0, 0.10. Mandibles: L 1234 0 a(m$^{1,2,3}$)b / R 1234 y a(m$^{1,2or1,2,3}$)b, in both mandibles tooth a > b. Prementum yellow, subquadrate, slightly wider than long, lateral margins smooth and slightly sinuate. Ligula (Fig. 5C) with a ventral row of 6–7 (usually 7) reddish-brown, short, truncate teeth on middle, and a dorsal row of short, stout piliform setae; apical lobe of labial palp stout, tip acute and finely serrate, internal margin concave with 9–11 large, truncate teeth, the 2–3 most apical teeth acutely pointed, external margin strongly convex and smooth (Fig. 5C); movable hook slightly shorter than palp.
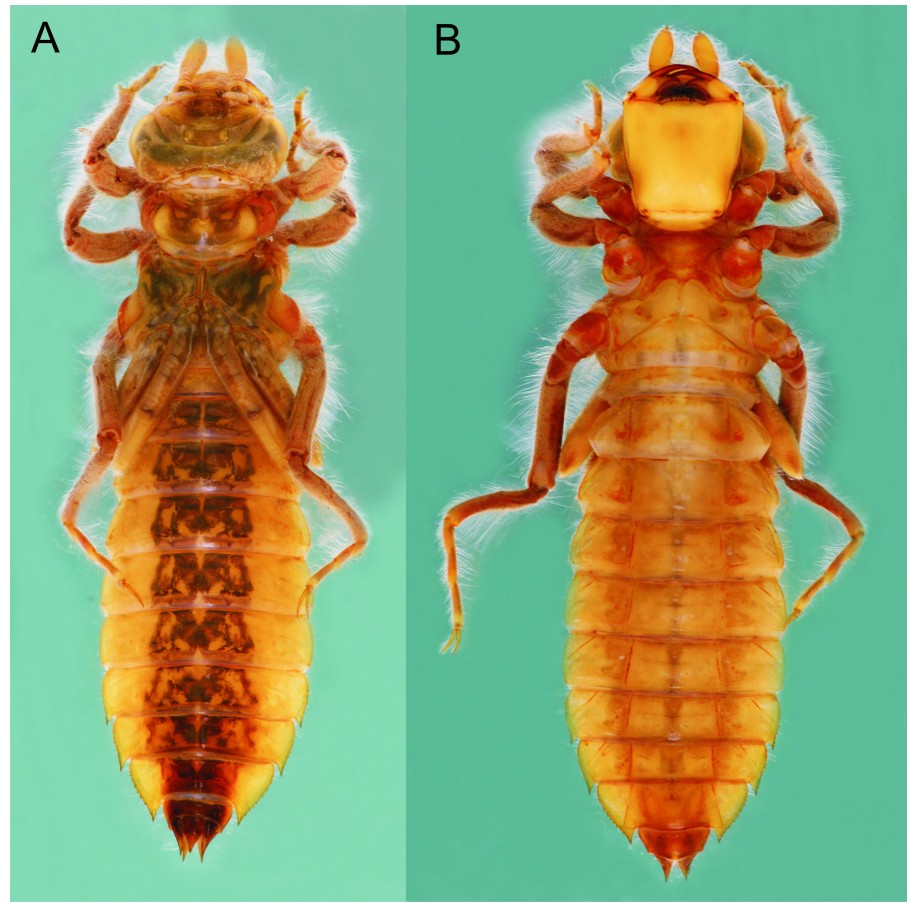

**Figure 8** *Epigomphus tumefactus*, last stadium, (A) dorsal view, (B) ventral view.

Thorax: As described for *E. jannyae* (Fig. 7).

Abdomen: As described for *E. jannyae* except: Yellowish-brown to light brown (Fig. 8), reddish-brown to dark brown on middle third of tergites 8–9, and the whole surface of S10; lateral margins of S1–6 with long, stiff, yellowish setae, shorter on S7–10, lateral margins of S6–9 serrate, serrations on S6 very small and only visible in lateral and ventral view (absent in one exuvia), serrations on S7–9 short and acute (Figs. 6C and 8), light brown to brown, S7–9 ending in sharply pointed spines, divergent on S7–8, parallel on S9 and increasing in size and robustness posteriorly (Fig. 6C), those on S9 slightly upturned. Caudal appendages pyramidal (Fig. 6C), sharply pointed, reddish-brown to dark brown, subequal in length; male's epiproct with a pair of dorsal tubercles at basal 0.50, widely convex apically and divergent (Fig. 6C); tips of cerci slightly divergent, tips of paraprocts parallel, dorsal and ventral margin of paraprocts carinated.

**Measurements.** Exuviae (*N* = 2): Total length 27.4–29.6; abdomen 16.3–17.7; hind femur 4.8–4.9; maximum width of abdomen (ventral) 6.9–7.2; caudal appendages 1.3–1.4; lateral spines on S7 0.4, on S8 0.6, on S9 0.6–0.7. Larvae (*N* = 2): Total length 26.7–28.6; abdomen 17.4–17.8; maximum width of head 5.6–6.0; hind femur 4.9; maximum width of abdomen

(ventral) 7.2–7.5; caudal appendages 1.3; lateral spines on S7 0.4–0.5, on S8 0.6–0.7, S9 0.7–0.8.

**Habitat.** *E. tumefactus* larvae were collected in 1st order streams draining mature forest. Larvae emerged on top of flat rocks on the channel margin.

## DISCUSSION

The larva of *Epigomphus jannyae* is similar to larvae of other described Epigomphus species, although they can be separated by the following features (in parentheses those of other species, including *E. tumefactus* above described): Integument abundantly granular (not granular but covered with minute, scale-like setae, *E. crepidus*, *E. echeverrii*, *E. subobtusus*, *E. subsimilis*); third antennomere claviform, 1.6 times longer than its widest part (spindle-shaped, 2 times longer or more than its widest part, *E. crepidus*, *E. echeverrii*, *E. subobtusus*, *E. subsimilis*, *E. tumefactus*); third antennomere with small, regularly spaced, circular structures close to borders (lacking such structures, *E. crepidus*); lateral margins of prementum slightly sinuate at apical 0.60 (lateral margins slightly convex, *E. crepidus*, *E. echeverrii*, *E. subobtusus*, *E. subsimilis*; straight and parallel at apical half, *E. hylaeus*); tip of palpal lobe rounded and smooth (tip of palpal lobe rounded and finely serrate, *E. crepidus*, *E. echeverrii*, *E. subobtusus*, *E. subsimilis*; tip of labial palp acute and finely serrate, *E. tumefactus*); movable hook almost as long as palp (as long as palp, *E. hylaeus*, *E. subsimilis*); lateral margins of S5 finely serrate (not serrated, *E. crepidus*, *E. hylaeus*, *E. echeverrii*, *E. subobtusus*, *E. subsimilis*, *E. tumefactus*); lateral margins of S6 serrate (no serrate, *E. crepidus*, *E. hylaeus*, *E. echeverrii*, *E. subobtusus*, *E. subsimilis*); lateral spines on S6–9 obtusely pointed (lateral spines on S7–9 sharply pointed, all other species); extreme tips of cerci suddenly and strongly out-turned (extreme tips of cerci slightly divergent, all other species).

The shape and length/width ratio of the 3rd antennomere and the rounded, smooth tip of the labial palp makes the larva of *E. jannyae* most similar to that of *E. hylaeus*. However, they differ by the larger stature of *E. jannyae* as well as the presence of serrations on the lateral margins of S5–9 and the obtusely pointed lateral spines on S6–9 of this species. Likewise, by the shape and length/width ratio of the 3rd antennomere, the larva of *E. tumefactus* appears related to *crepidus, echeverrii, subobtusus,* and *subsimilis,* differing from all these species by the acute tips of its labial palps. In this last feature it could be related to *E. paludosus*, according with the drawing provided by *Martins (1968)*.

The descriptions here provided would shed light on the relationship of *Epigomphus* with other genera of Gomphidae. Several authors have tried to relate *Epigomphus* to other genera based mainly on adult morphology (e.g., *Williamson, 1920*), while others have included some larval characters (*Carle, 1986*; *Belle, 1996*). According to our present knowledge of the larvae, we think that *Carle*'s (*1986*) classification best reflects the relationships among the genera that he placed in the subfamily Epigomphinae, excepting the Macrogomphini (*Macrogomphus* only), whose larvae more resemble those of Carl's tribe Gomphoidini. *Epigomphus* is probably closely related to the oriental *Leptogomphus* (tribe Leptogomphini) by the following combination of features: antennae with abundant scale-like setae, dual

shape of 3rd antennomere; molar lobe of right mandible with formula $a(m^{1-2})b$; shape of prementum, ligula poorly developed and slightly convex; lateral margins of S6–9 serrate, with lateral spines on S7–9; anal pyramid short; sternum 3 divided in 5 sternites; the last 3 features are also shared with the larvae of *Heliogomphus*, which also belongs to the Leptogomphini (*sensu Carle, 1986*). However, there are two striking differences between *Leptogomphus* and *Epigomphus*—the presence in the former of a ventral, longitudinal, median sulcus on prementum, and abdominal dorsal protuberances.

The larvae of *E. jannyae* and *E. tumefactus* would fit in the key provided by *Novelo-Gutiérrez, Gómez-Anaya & Smith-Gómez (2015)*, after the following modifications:

1. Third antennomere claviform (length/width proportion l.6:1); tip of labial palp rounded and smooth..................................................................................................... 2.

1'. Third antennomere spindle-shaped (length/width proportion $\geq$ 2:1); tip of labial palp rounded or acute and finely serrate................................................................. 3.

2. Lateral margins of prementum slightly sinuate; lateral margins of S5–9 serrate; lateral spines on S6–9 triangular, obtusely pointed (very small on S6, visible only in ventral view); extreme tips of cerci suddenly and strongly out-turned; total length more than 23 mm................................................................................................................*jannyae*.

2'. Lateral margins of prementum straight and parallel on apical half; lateral margins of S7–9 serrate; lateral spines on S7–9 sharply pointed; tips of cerci slightly divergent; total length less than 23 mm ................................................................................ *hylaeus*.

3. Tip of labial palp acute; ligula with 6–7 teeth; lateral margins of S6 serrate......................
..................................................................................................................*tumefactus*.

3'. Tip of labial palp rounded; ligula with 8 or more teeth; lateral margins of S6 no serrate.................................................................................. 4 (continues in original key).

## CONCLUSION

*Epigomphus* continues to be a poorly known group, only eight of the 28 species are known as larvae. However, descriptions here provided and comparisons with the literature allow for the assessment of important larval characters useful for the separation of species.

## ACKNOWLEDGEMENTS

We thank M. Sc Aydeé Cornejo and the Instituto Conmemorativo Gorgas de Estudios de la Salud for facilitating collecting trips in Panama. The manuscript was also improved by valuable comments provided by reviewers.

### Funding

The Instituto Conmemorativo Gorgas de Estudios de la Salud, Panama, facilitated the collecting trips in Panama. The funders had no role in study design, data collection and analysis, decision to publish, or preparation of the manuscript.

## Grant Disclosures

The following grant information was disclosed by the authors:
Instituto Conmemorativo Gorgas de Estudios de la Salud, Panama.

## Competing Interests

The authors declare there are no competing interests.

## Author Contributions

- Rodolfo Novelo-Gutiérrez conceived and designed the experiments, performed the experiments, analyzed the data, contributed reagents/materials/analysis tools, wrote the paper, prepared figures and/or tables, reviewed drafts of the paper.
- Alonso Ramírez and Débora Delgado conceived and designed the experiments, performed the experiments, analyzed the data, contributed reagents/materials/analysis tools, wrote the paper, reviewed drafts of the paper.

## Data Availability

Specimens studied are deposited in Colección Entomológica del Instituto de Ecología, A.C., Xalapa (IEXA) and the Museo de Zoologia, Universidad de Cosa Rica (MZUCR).

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
