# Peer review of "The larvae of Epigomphus jannyae Belle, 1993 and E. tumefactus Calvert, 1903 (Insecta: Odonata: Gomphidae)"

_PeerJ, doi:10.7717/peerj.2338_

## Round 0.1 · original submission · Minor Revisions

All three expert reviewers suggest minor revisions and provide important points of consideration in their written reviews and marked-up PDFs. The authors should consider and respond to all of these comments and provide a comprehensive rebuttal letter outlining the changes that they have made and the reasons (if applicable) for not making certain suggested changes.

·

Basic reporting

The article is clear and unambiguous; the introduction and the background demostrate is the appropiate for this field, in addition is appropriately referenced. This paper is structured according to the the discipline-specific custom.
Figures are relevant, nevertheless, there are some comments on its legends.
I suggest to the authors to include basic information on the geographic distribution of both species of Epigomphus.

Experimental design

This contribution describe original primary research, which is relevant. Methods are adequate and properly described.

Validity of the findings

Results and conclusions are robust and well sustained

Additional comments

I suggest to include basic information on the geographic distribution of both species of Epigomphus. Other comments are on the manuscript

·

Basic reporting

This is a good, solid treatment of new information on the larvae of an interesting and important group of Odonata. Detailed descriptions of odonate larvae are vital to our understanding of the ecology, conservation and phylogeny of these important insects. The material is original and is placed in useful context. The relevant literature is cited. In general, the text is well-written, with only a few instances where I think syntax and /or clarity could be improved. I have indicated these directly on the manuscript. In particular, the punctuation of the text of the larval descriptions should be examined for clarity. In the ms (first species description only), I have changed many commas to semi-colons in an attempt to clarify the descriptions. The authors should check these to make sure that the details are easily read and are clearly understood; they should then examine the punctuation of the description of the second species and modify where necessary.

Figures are clear and relevant to the identification of the two species, which is the main purpose of the article. The caption for Fig. 6 is missing and needs to be added (see ms). Although I do not insist on it, the inclusion of a couple of figures showing the habitat where larvae and exuviae were collected would add to the usefulness of the paper.

The modified portion of the identification key to Epigomphus is clear and straightforward, and the distinguishing characters treated in the diagnosis (discussion) and used in the key are useful and readily seen and measured. This is critical for the utility of such diagnostic tools.

Experimental design

The data and results of this research are original. The relatively poor understanding we have of the larval stages of Epigomphus species and their relationships within, and outside, the genus are stated. The knowledge gap is identified and discussed relative to the larvae being described. The larval descriptions and diagnostic methods are of good quality and can be checked and reproduced by other workers. The specimen material examined has been deposited in recognized collections and is available for future use.

Validity of the findings

Results are clear and useful. Although the sample size of specimens examined is very small, it is sufficient in the context of distinguishing the species in question. The presentation of this new data enabling identification of two more species of Epigomphus is the purpose of the research. As noted above, the specimen material examined and described has been deposited in recognized collections and is available for future use.
The conclusions are not overstated and encourage further examination of the questions raised.

Additional comments

No additional comments. I'd be happy to have the comments in the other boxes (and on the manuscript) sent directly to the authors.

·

Basic reporting

Please, see "General Comments for the Author"

Experimental design

Please, see "General Comments for the Author"

Validity of the findings

Please, see "General Comments for the Author"

Additional comments

Dear Editor and authors

Thanks for invite me for reviewing this manuscript. I hope had contributed to improve it, as well as with the quality of PeerJ. Since I have not kept myself anonymous as reviewer in my commentaries along the text I do not have any objection in sharing this message with the author. Feel free to forward all commentaries listed below to him. The manuscript is worthwhile to be published and my decision is suggest for publication pending modifications by the authors. My suggestions/corrections are directly in the PDF file, I will highlight only the major points here.

1. Overall commentary. The manuscript represents a very simple and straightforward study with description of two hitherto unknown larvae of clubtail dragonflies (Gomphidae), although a welcome contribution about these insects from tropical America, adding an important piece to the knowledge of this neglected biota. The descriptions were well-conducted and the key is workable; however some sentences must be improved. The photos, although are good enough, they should be rearranged into the plates, they are with unequal sizes and unaligned. My major concern relies on the discussion section in which some key aspects were overlooked and it provides few misleading data. The first paragraph of the "Discussion" is more like a diagnosis and corresponds fairly well to what several journals prefer adopt as a “Remarks” section. I suggest reorganize its structure, with the "Discussion" with a subheading similarly to Diagnosis, Remarks, Taxonomy notes etc. This section can be significantly improved. Furthermore, in my opinion the section “Conclusion” is strongly unnecessary instead it can be the last sentence of the discussion section.

2. Title: It could be improved to be more precise. Please, check suggestion did directly on the file.

3. Abstract: Adequate, suggestions aiming precision and standardization are on the file.

4. Introduction: Concise and somewhat reductionist, but adequate, few sentences require citations and corrections, see PDF file.

5. Discussion. See commentaries on the file.

6. Figures and captions: The figures are under a good quality, but would be welcome standardize the size, position, alignment etc. Captions of figures can be strongly improved and modified to be in agreement with journal style, I did minor suggestions aiming standardization.

All other corrections/suggestions are directly in the PDF file using tracking changes and commentaries (highlighted in yellow).

Sincerely,
Ângelo P. Pinto

---

## Round 0.2 · accepted · Accept

The authors have completed or rebutted all of the suggested minor reviviojs adequately, and this paper is now ready for publication in PeerJ.

I encourage the authors to consider making the review history of this MS public, as it adds value to the overall submission, and as all three reviewers opted to make their identities known.